# Contrasting late-glacial paleoceanographic evolution between the upper and lower continental slope of the western South Atlantic

Leticia G. Luz[1], Thiago P. Santos[2], Timothy I. Eglinton[3], Daniel Montluçon[3], Blanca Ausin[3,4], Negar Haghipour[3], Silvia M. Sousa[4], Renata H. Nagai[5], Renato S. Carreira[1]

[1] LabMAM/Departamento de Química, Pontifícia Universidade Católica do Rio de Janeiro (PUC-Rio), Rio de Janeiro, Brasil
[2] Programa de Geociências (Geoquímica), Universidade Federal Fluminense, Niterói, Brazil
[3] Department of Earth Science, Geological Institute, ETH Zürich, Zürich, Switzerland
[4] Department of Geology, University of Salamanca, Salamanca, Spain
[5] Instituto de Oceanografia, Universidade de São Paulo, São Paulo, Brasil
[6] Centro de Estudos do Mar, Universidade Federal do Paraná (UFPR), Paraná, Brasil

*Correspondence to*: Leticia G. Luz (leticiagluz@gmail.com)

**Abstract.** The number of sedimentary records collected along the Brazilian Continental Margin has increased significantly in recent years, but relatively few are located in shallow waters and register paleoceanographic processes in the outer shelf-middle slope prior to 10-15 ka BP. For instance, the northward flow up to 23-24 ºS of cold and fresh shelf waters sourced from the subantarctic region is an important feature of current hydrodynamics in the subtropical western South Atlantic Ocean, and yet limited information is available for the long-term changes of this system. Herein, we considered a suite of organic and inorganic proxies – alkenones-derived sea surface temperature (SST), $\delta$D-alkenones, $\delta^{18}$O of planktonic foraminifera and ice-volume free seawater $\delta^{18}O_{IVF-SW}$ – in sediment from two cores (RJ-1501 and RJ-1502) collected off the Rio de Janeiro shelf (SE Brazilian continental shelf) to shed light on SST patterns and relative salinity variations since the end of the last glacial cycle in the region and the implications of these processes over a broader spatial scale. The data indicate that, despite the proximity (~ 40 km apart) of both cores, apparently contradictory climatic evolution occurred at the two sites, with the shallower (deeper) core RJ-1501 (RJ-1502) showing consistently cold (warm) and fresh (salt) conditions toward the Last Glacial Maximum and last deglaciation. This can be reconciled by considering that the RJ-1501 core registered a signal from mid- to high-latitudes on the upper slope off Rio de Janeiro represented by the influence of the cold and fresh waters composed of Subantarctic Shelf Water and La Plata Plume Water transported northward by the Brazilian Coastal Current (BCC). The data from core RJ-1502 and previous information for deep-cores from the same region support this interpretation. In addition, alkenone-derived SST and $\delta^{18}O_{IVF-SW}$ suggest a steep thermal and density gradient formed between the BCC and Brazil Current (BC) during the last climate transition which, in turn, may have generated perturbations in the air-sea heat flux with consequences for the regional climate of SE South America. In a scenario of future weakening of the Atlantic Meridional Overturning Circulation, the reconstructed gradient may become a prominent feature of the region.

# 1 Introduction

The paleoclimatic knowledge accessed through marine sediment cores from the extensive Brazilian continental margin has increased significantly through the last decades (e.g., Arz et al., 1999, 2001; Chiessi et al., 2008; Govin et al., 2014; Jaeschke et al., 2007; Jennerjahn et al., 2004; Lessa et al., 2017; Mulitza et al., 2017; Portilho-Ramos et al., 2015; Rühlemann et al., 1999). The records distributed along the western South Atlantic display considerably heterogeneous features in terms of sedimentation rate and response to the adjacent continental climate, which makes the scientific outputs extremely dependent on the core site location. Those cores located adjacent to the semi-arid NE Brazil allowed several investigations addressing the interplay between changes in the Atlantic Meridional Overturning Circulation (AMOC), the sea surface temperature (SST) of the tropical Atlantic, and the continental climate (in terms of precipitation and vegetation cover) in centennial to millennial time-scales (Behling et al., 2000; Bouimetarhan et al., 2018; Burckel et al., 2015; Crivellari et al., 2019; Venancio et al., 2018; Zhang et al., 2015, 2017). Otherwise, marine records recovered from the subtropical realm (southern to 20 ºS) generally do not show the obvious sea surface millennial-scale features like those from NE Brazil (Santos et al., 2017a); nonetheless, this area suffers large changes in the wind-driven upwelling pattern with consequences for regional upper-ocean productivity (Lessa et al., 2019; Portilho-Ramos et al., 2019). The subtropical Brazilian margin might also be a sensitive region to the transmission of the Agulhas rings to the western South Atlantic at the end of glacial periods, highlighting its importance for glacial-interglacial transitions (Santos et al., 2017b; Ballalai et al., 2019). Cores from the subtropical margin also have been used to explore changes in the water mass composition of the deep Atlantic during the last glacial cycle (Howe et al., 2018; Lund et al., 2015; Oppo et al., 2015; Tessin and Lund, 2013).

Most of these studies were based on intermediate to deep-water cores (> 1500 m) recovered from the mid- to lower-continental slope, and investigations supported by shallower water-depth cores (< 1000 m) are generally limited to the Holocene (e.g., Albuquerque et al., 2016; Lessa et al., 2016; Nagai et al., 2014). Such a shorter-time range from shallower-water cores is due to the fact that wide regions of the continental shelf were exposed during the last glacial cycle, and that the sea-level rise of the last deglaciation provoked massive sedimentological disturbances, preventing the acquisition of well-organized chronological sequences. This hindrance was partially circumvented by cores GeoB2107-3 and GeoB6211-2, retrieved at 1048 m and 657 m, respectively, in which dinocyst assemblage reconstructions were developed (Gu et al., 2017; Gu et al., 2018). These authors verified the presence of eutrophic taxa along the SE Brazilian coast from the late glacial to the last deglaciation and assigned this to the high input of continental nutrients carried by the Brazilian Coastal Current (BCC) that flowed farther from the shore due to the low sea-level. However, a more straightforward hydrographic reconstruction of the eventual influence of the BCC along the southern portion of the Brazilian continental margin is still lacking. This limits the elaboration of detailed knowledge about the regional climatic evolution, mainly for inner parts of continental margin that might receive the influence of the BCC.

The BCC is a seasonal coastal current flowing northward at the continental shelf from the Argentinean shelf and southern Brazilian shelf that transport a mixture of cold and low-salinity Plata Plume Water (generated by the discharge of the La Plata

River) and Subantarctic Shelf Water (derived from the northern extension of the Malvinas Current) (Möller et al., 2008; de Souza and Robinson, 2004). The BCC is forced by the prevailing southern winds allied to the meridional oscillations of the Brazil-Malvinas Confluence (Stevenson et al., 1998). The cold and low-salinity BCC exhibits very contrasting hydrographic patterns compared to those of the warm and high-salinity Brazil Current (BC) (Mendonça et al., 2017). As a consequence, the presence of the BCC over the continental shelf produces a cross-shelf SST gradient of ~ 1.5 ºC toward the shelf break, where BCC waters meet the opposite flow of the BC. The gradient imposed by this thermal front may disturb atmospheric properties, such as surface wind stress, stability, and air-sea flux exchange, as the sea surface in this region can act as a heat source to the atmosphere (Pezzi et al., 2016). Simulations using the Regional Climate Model (RegCM3) suggested that air-sea moisture and heat exchanges play a role in controlling the annual cycle of precipitation over the southern areas of eastern South America (Reboita et al., 2010) and projected scenarios of increased precipitation, mainly in summer and spring, in both the near (2010-2014) and distant (2070-2100) future (Reboita et al., 2014). Furthermore, the northward spreading of the Plata Plume and Subantarctic waters promotes the injection of nutrients and, consequently, high chlorophyll-a concentrations in the photic zone, increasing regional primary production (Ciotti et al., 1995). Hence, the dynamics of the BCC are a determining factor for the climate and ecosystem along the SE Brazilian coast.

In terms of climate reconstruction, sediment cores retrieved in areas under the influence of the BCC may not follow temperature and salinity patterns previously reconstructed for the BC (e g., Santos et al., 2017a). Moreover, the BCC carries a fingerprint of the continental climate, as most of the drainage systems in SE Brazil flow inland to the Paraná Basin and are discharged to the ocean by the La Plata River. This makes marine cores under the BCC sedimentation regime ideal in also reconstructing inland climate conditions from the continental runoff in the subtropical western South Atlantic. Herein, we present a paleoceanographic reconstruction based on two sediment cores collected in the upper (RJ-1501, 328 m water depth) and lower (RJ-1502, 1598 m water depth) continental slope using organic (alkenones-derived SST and $\delta$D-Alkenones) and inorganic ($\delta^{18}$O of planktonic foraminifera and ice-volume free seawater $\delta^{18}O_{IVF-SW}$) proxies for the last glacial transition aiming to reconstruct the BCC evolution during the last glacial-interglacial transition and placing it in the context of the offshore waters influenced by the BC. Our multi-proxy reconstruction indicates that RJ-1502 (the furthest record from the coast) agree relatively well with earlier studies developed in the BC core (Santos et al., 2017a), which reported a gradual build-up in the temperature and salinity along the end of the last glacial towards the Holocene without minimum SST during the Last Glacial Maximum (LGM). On the other hand, the shallower RJ-1501 revealed that the evolution of inner waters occurred in the opposite direction. This core indicates an accentuated cooling around the LGM and an increase in continental freshwater discharge during the last deglaciation. We interpret this antagonism as the result of the influence of the BCC and its cold low-salinity waters that carried (i) the temperature evolution pattern from the mid- to high- South Atlantic latitudes, and (ii) the enhanced precipitation signal in the adjacent SE South America during the last deglaciation to the study area. Our data, therefore, strengthen knowledge on the long-term circulation changes in this region and highlight the nonlinearity of the climate system even in neighbouring regions.

## 2 Material and Methods

### 2.1 Sediment cores

00        Two gravity cores were collected in the slope off southeast Brazilian Continental Margin by RV Inspector II in June 2015 (Figure 1). The RJ-1501 (23º58'14.3" S/43º06'35.1" W) is a 402 cm length core retrieved at the water depth of 328 m whereas the core RJ-1502 (24º32'57.6" S/42º55'42.9" W), retrieved at 1598 m water depth, has 450 cm length (only the data from the first 250 cm were considered due to $^{14}$C-dating limitation). Both cores were sliced at 3 cm intervals, resulting in 134 and 84 samples for cores RJ-1501 and RJ-1502, respectively. Samples were stored frozen in an aluminium container and 05  subsequently freeze-dried.

### 2.2 Regional settings

The current study was carried out with sediment cores collected from the upper and intermediate slopes of the state of Rio de Janeiro, Santos Basin, located in the subtropical western South Atlantic (Figure 1). The offshore circulation in the area is governed by the western portion of the anticyclonic movement of the South Atlantic Subtropical Gyre. The South Equatorial 10  Current reaches the Brazilian margin and is distributed into two surface flows, at approximately 10-15º S, namely: the North Brazil Current and the BC (Peterson and Stramma, 1991; Stramma and England, 1999). The BC flows south along the Brazilian margin with a total width of 400-500 m, carrying the nutrient-poor Tropical Water (TW: T>20°C; S>36) in its upper ~ 100 m and the nutrient-rich South Atlantic Central Water (SACW: T=6-20°C; S~34.6-36) between ~ 100 and 600 m (Silveira et al., 2017). At 33-38ºS, the southward flow of the BC meets the northward flow of the Malvinas Current to form the Brazil-Malvinas 15  Confluence (Figure 1). The position of the Brazil-Malvinas Confluence presents a strong seasonality, moving to a lower latitude (~ 34 ºS) with a northward penetration of the Malvinas Current during the austral winter and a higher latitude (~ 40 ºS) with a southward penetration of the BC during the austral summer (Olson et al., 1988). Both the BC and Malvinas Current are deflected eastward at the Brazil-Malvinas Confluence region, feeding the South Atlantic Current.

Over the continental shelf of the subtropical western South Atlantic, water masses are originated by the dilution of open ocean 20  waters from the western boundary currents. Two distinct water masses are identified, the cold and low-salinity Subantarctic Shelf Water and the warm and high-salinity Subtropical Shelf Water (Piola et al., 2008). The origin of the Subantarctic Shelf Water is related to precipitation excess and continental runoff in the southeast Pacific that penetrates the South Atlantic south of Cape Horn, flowing northward. The Subtropical Shelf Water is fed by the detachment of the TW from the surface layer of the BC. At ~ 33 ºS, a narrow frontal zone, referred to as the Subtropical Shelf Front, separates the two water masses (Piola et 25  al., 2000). A cross-shelf section indicates no penetration of the Subantarctic Shelf Water north of Subtropical Shelf Front, yet hydrographic and satellite observations indicate that cold (14-17 ºC) and low-salinity (33.0-34.0) water tongues can be traced to latitudes as low as 23 ºS (Campos et al., 1996). This low-SST and salinity water observed beyond the front is associated with the northward spreading of the La Plata River outflow (Möller et al., 2008). The discharge of the so-called Plata Plume Water produces a major impact on vertical stratification, since northward penetration of the river plume is associated with

decreased surface salinity (Palma et al., 2008; Piola et al., 2005). Along the southern Brazilian margin, the northward movements of the Subantarctic Shelf Water and Plata Plume Water are determined by the BCC. The BCC is a wintertime coastal current flowing northward on the continental shelf forced by the prevailing southern winds of the winter and meridional displacement of the Brazil-Malvinas Confluence (de Souza and Robinson, 2004) (Figure 1). The BCC, therefore, carries cold and low-salinity waters northward and flows opposite to the BC, which carries warm and salty waters southward (Figure 1). The shearing between these two currents produces an intense across-shore thermal gradient in which mass and heat exchanges occur via turbulent mixing along their boundaries (Mendonça et al., 2017; Pezzi et al., 2016).

## 2.3 Age model

The RJ-1501 and RJ-1502 core chronologies were obtained through AMS $^{14}$C dating over the shells of the surface-dwelling planktic foraminifera *Globigerinoides ruber* and *Trilobatus sacculifer*. The AMS $^{14}$C was measured in ten and eight samples of core RJ-1501 and RJ-1502, respectively (Table 1). Each sample comprised roughly 10 cm³ of sediment, and 50 shells of the mentioned species were handpicked using a stereomicroscope from the 250 µm size-fraction. AMS $^{14}$C ages were determined at the ETH Laboratory of Ion Beam Physics (Zurich) using the mini radiocarbon dating System (MICADAS). This system is based on a vacuum insulated acceleration unit that uses a commercially available 200 kV power supply to generate acceleration fields in a tandem configuration. This technique is capable of determining low $^{14}$C concentrations due to the high energies employed in the particle accelerator and the magnetic and electrostatic mass analyzers (Synal et al., 2007).

The age-depth model was built using the Bacon v. 2.3 software, which uses Bayesian statistics to reconstruct Bayesian accumulation histories for sedimentary deposits (Blaauw and Christeny, 2011). The $^{14}$Cages were calibrated using the IntCal13 curve (Reimer et al., 2013) and modelled with a reservoir age of 375 ± 36 years from ten local records (Figure 2). BACON was run with default parameter settings, except for a higher memory (mem.mean = 0.7) to consider that the sedimentation rate of a particular depth in the cores depends on the depth above it. Ages were modeled using a student-t distribution, with 33 degrees of freedom (t.a=33, t.b=34) and 10,000 age-depth realizations to estimate median age and 95% confidence intervals at 3 cm resolutions for each core. Mean 95% confidence ranges were of 2588 and 3861years for cores RJ-1501 and RJ-1502, respectively. One hundred percent off the dates lie within the age-depth model in a 95% range for both cores. According to our age-depth models, cores RJ-1501 and RJ-1502 cover the last 42.44 and 53.57 ka BP (Figure 2). The mean sedimentation rate of core RJ-1501 was around 100 years/cm between 40 and 20 ka BP, with a steep increase to 500 years/cm during the last deglaciation and a return to previous values during the Holocene. A general lower sedimentation rate was found for core RJ-1502, with roughly 200 years/cm between 50 and 20 ka and almost 1000 years/cm during the last deglaciation and Holocene (Figure 2).

## 2.4 Alkenone analyses and sea surface temperature reconstruction ($U_{37}^{K'}$-derived SST)

A total of 139 sediment samples were selected for alkenone analyses, considering the two collected cores. Sampes were freeze dried and homogenized with a mortar and pestle. An 15-30 g (± 0.1 mg) aliquot of the homogenized material was extracted in a pressurized solvent extractor system (Dionex® ASE-200) using a mixture of dichloromethane (DCM): methanol 9:1 (v/v) at 100 °C and 1000 psi in two cycles, 11 minutes per cycle. Before extractions, a known amount of 2-nonadecanone was added as a surrogate standard. The total lipid extracts were saponified (1M KOH at 110 °C for 2h), and the neutral fraction recovered with $n$-hexane. The neutrals were further fractionated using a Pasteur pipette containing 4 cm of activated silica into three fractions: apolar (4 mL of $n$-hexane), ketones (4 mL of $n$-hexane:DCM, 2:1/v:v) and polar (4 mL of DCM:methanol, 1:1/v:v). The alkenone analyses were performed at the Chemistry Department from PUC-Rio. The alkenones in the ketones fraction were identified and quantified using a Thermo® Focus gas chromatograph equipped with an Agilent DB-5 capillary column (60 m x 250 µm diameter x 0.25 µm internal film) and a flame ionization detector. The oven temperature program used started at 50 ºC, followed by a 20 °C min⁻¹ ramp up to 80 °C and a second ramp at 8 °C min⁻¹ up to 320 °C, with a final hold at this temperature for 33 min. The extracts (1 µL) were injected in spitless mode and He was used as the carrier gas at 1.2 mL min⁻¹. Alkenone identification was based on the retention time of authentic standards, whereas quantification followed the internal standard method using $n$-$C_{36}$ alkane. Analytical precision was estimated as 12% or better, based on triplicate analyses of a sediment sample. The $U_{37}^{K'}$ values were calculated according to Prahl and Wakeham (1987): $U_{37}^{K'} = C_{37:2}/(C_{37:2} + C_{37:3})$ and were then converted to sea surface temperature ($U_{37}^{K'}$-derived SST) using the Müller et al. (1998) calibration, as follows: $U_{37}^{K'} = 0.033$ $U_{37}^{K'}$-derived SST (ºC) + 0,069; $r^2$ = 0.98; n = 149; sd = ± 1.0 ºC). This equation was chosen because it is derived from 149 tropical to subpolar eastern South Atlantic surface sediments and was already used in other studies from the same region (e.g., Ceccopieri et al., 2018).

## 2.5 Planktic foraminifera oxygen ($\delta^{18}O$) and carbon ($\delta^{13}C$) isotopic compositions

The oxygen ($\delta^{18}O$) and carbon ($\delta^{13}C$) isotopic compositions were determined in planktic foraminifera *Globigerinoides ruber* [white] shells from selected slices of the RJ-1501 and RJ-1502 cores. Approximately 10 mL of sediment were sifted sequentially using two sieves (63 µm and 150 µm meshes), hand-picking the specimens. Approximately 10-15 shells >150 µm from each sample were weighed in appropriate glass vials to be injected into an automatic Kiel IV Thermo Fisher Scientific® system (automatic $CO_2$ obtainment device from the analyzed carbonate, Geological Institute, ETH-Zurich) for the oxygen and carbon isotopic analyses. The determination of low ¹⁸O and ¹³C isotope contributions is performed by using the respective ratios with their most abundant elements in the sample. The data were reported by the delta notation in parts per thousand and relative to the standard Vienna Pee-Dee Belemnite (VPDB). The Kiel IV carbonate device was coupled to a Thermo Fisher Scientific® Delta V Plus mass spectrometer. The carbonate generated by the shells was vacuum dissolved by applying a phosphoric acid 103% drip at a temperature of 70 ºC and directed to the mass spectrometer. The masses were calibrated with the international MS2 ($\delta^{18}O_{VPDB}$ = 1.81‰, $\delta^{13}C_{VPDB}$ = 2.13‰, n = 14) and ISOLAB B ($\delta^{18}O_{VPDB}$ = -18.59‰, $\delta^{13}C_{VPDB}$ =

10.20‰, n = 4) standards and all results were reported as the conventional delta notation in relation to the VPDB standard. The standard deviations were $\delta^{18}O = 0.061$ and $\delta^{13}C = 0.040$ for the MS2 standard and $\delta^{18}O = 0.076$ and $\delta^{13}C = 0.076$ for the ISOLAB B standard.

## 2.6 Ocean surface salinity tracers (Alkenone hydrogen isotope and Ice-volume free seawater oxygen isotope)

95 The 2H/1H isotopic ratios of the C37:2-3 alkenones (δD-Alkenones) produced by haptophyte algae was used as a proxy for changes in sea surface salinity (SSS). The F2 fraction separated from the organic lipid extract (see section 2.4) was used to determine the δD-Alkenones by gas chromatography-isotopic ratio mass spectrometry (GC-IRMS; Thermo TraceGC gas chromatography coupled to a Thermo Delta V Plus mass spectrometer) at the Geological Institute (ETH-Zurich). A relative large volume (up to 8 µL) of the extract, corresponding from 50 to 300 ng of the analyte, was injected in a PTV inlet (Gerstel,

00 CIS-6) set to the solvent-vent mode for a better quantitative transfer onto an Agilent VF-1ms column (60 m x 0.25 mm x 0.25 µm). The oven temperature programme was set to have the alkenones eluting isothermally at 300ºC in order to minimize column bleeding. Prior to each sample injection, the high temperature pyrolysis reactor was conditioned with a 4 s pulse of methane through the reactor to maintain the reactor's pyrolysis efficiency (Cao et al., 2012). A standard of n-C27 alkane (Heptacosane#3, δ2HVSMOW = -172.8‰ ± 1.6‰, from Arnd Schimmelmann, Biogeochemical Laboratory, Indiana

05 University) was used to construct a calibration curve with a linear response of the IRMS signal up to 100 V. For more intense detector signals (> 100 V), a constant calibration offset was used. Propagated errors on duplicate analysis varied between 2 and 8‰.

To determine the sea water oxygen isotope ($\delta^{18}O_{sw}$), we used the *G. ruber* [white] $\delta^{18}O$ values and $U_{37}^{K'}$-derived SST applying the equation: $T (U_{37}^{K'}$-derived SST$) = -4.44 * (\delta^{18}O - \delta^{18}O_{IVF-SW}) + 14.20$ (Mulitza et al., 2003)). The effect of global sea-level

changes was subtracted from $\delta^{18}O_{sw}$ to generate an ice-volume free seawater $\delta^{18}O$ ($\delta^{18}O_{IVF-SW}$) based on the sea-level reconstruction of Grant et al. (2012) and considering a glacial $\delta^{18}O$ increment of 0.008 per meter sea-level lowering (Schrag et al., 2002; Simon et al., 2013). A factor of 0.27 ‰ was applied to convert the values from VPDB to Vienna Standard Mean Ocean Water (VSMOW) (Hut, 1987). $\delta^{18}O_{IVF-SW}$ uncertainty considers the uncertainty in $U_{37}^{K'}$-derived SST of ± 1.0 ºC – which is equivalent to a 0.22‰ $\delta^{18}O$ change (Mulitza et al., 2003) – and a 0.06 ‰ analytical error for calcite $\delta^{18}O$ measurement. The

total propagated cumulative uncertainty was ± 0.20 ‰, consistent with earlier studies that applied this proxy in the vicinity of our study area (e.g., Chiessi et al., 2015). Previous publications have used the combination of $U_{37}^{K'}$-derived SST and foraminifera $\delta^{18}O$ to estimate the $\delta^{18}O_{IVF-SW}$. In these publications it is generally assumed that the dominant alkenone producer *Emiliania huxleyi* and *G. ruber* have comparable water depth habitats (e.g., Rostek et al., 1993; Emeis et al., 2000; Carter et al., 2008; Sepulcre et al., 2011; Kasper et al., 2014), which is also the case in the subtropical western South Atlantic (Venancio

et al., 2017; Ceccopieri et al., 2018). Seasonal corrections over the $U_{37}^{K'}$-derived SST before application in $\delta^{18}O_{IVF-SW}$ has been used in regions of extreme seasonal variations in temperature and salinity, as the Mediterranean Sea (e.g., Essallami et al., 2007), which is not the case of the subtropical western South Atlantic.

## 3 Results

As a general result a same trend was noted for $\delta^{18}O$-, paleotemperature and relative changes in salinity proxies over the length of cores RJ-1501 and RJ-1502 (Figure 3A-E). Both records indicate similarity throughout MIS3 and MIS1 intervals. Regarding the slope variations (RJ-1501 → RJ-1502), $\delta^{18}O$ (Figure 3A) and $U^{K'}_{37}$-derived SST (Figure 3C) profiles are in agreement throughout the glacial period until Termination I, when a decoupling occurs and RJ-1501 data display lower temperature results. From the beginning of MIS 2, a discrepancy between the records indicates a genuine change in behaviour by the main environmental conditions. Unlike $\delta^{18}O$ and $U^{K'}_{37}$-derived SST profiles, a trend of convergence between the cores $\delta^{13}C$ data (Figure 3B) can be observed in MIS2, particularly during the last deglaciation. The isotopic composition of *G. ruber* ranged from -0.97‰ to 0.72‰ ($\delta^{18}O$) and 0.12‰ to 1.65‰ ($\delta^{13}C$) for the 101 considering the 101 samples for the RJ-1501 core and from -0.84‰ to 0.95‰ ($\delta^{18}O$) and 0.27‰ to 1.45‰ ($\delta^{13}C$) for the 72 samples for the core RJ-1502. A well-defined decreasing trend of $\delta^{18}O$ data is observed after the start of the LGM in the core RJ-1501 – although less evident in the deeper core RJ1502 – and is accompanied by an increase in $\delta^{13}C$ values from ~ ca. 12-13 ka in the two cores. The mean $U^{K'}_{37}$-derived SST was 21.8 ± 2.5 °C (n = 77) for core RJ-1501 and 20.7 ± 1.7 °C (n = 70) for core RJ-1502. $U^{K'}_{37}$-derived SST varied by 8.2 °C (17.6-25.8 °C) at the location closest to the shelf break and by 8.9 °C (16.9-25.8 °C) at the slope, with common positive bias located at ca. 39 ka and from ca. 12 ka until recently. Nevertheless, the modern annual SST means are higher +0.74 °C (RJ-1501) and +1.19 °C (RJ-1502) (data from World Ocean Atlas 2013 - WOA13, Locarnini et al., 2013) compared to our Holocene sea surface temperature estimates (RJ-1501 = 24.6 ± 0.88 °C and RJ-1502 = 24.8 ± 0.84 °C).

The relative changes in the sea surface salinity interval obtained by RJ-1501 and RJ-1502 cores data include the late MIS3 to MIS1. Despite the inconsistency of the $\delta^{18}O_{IVF-SW}$ and δD-Alkenone patterns between ca. 28 and 25 ka, both relative changes in salinity parameters display a good agreement and a change in the LGM and last deglaciation separating the local conditions of cores RJ-1501 and RJ-1502 is clearly observed (Figure 3 D-E). The $\delta^{18}O_{IVF-SW}$ records for RJ-1501 and RJ-1502 are presented in Figure 3D and reveal a millennial scale variability in the last 50 ka. The mean values of this relative salinity proxy was 1.56 (± 0.54‰, n = 63) at site RJ-1501 and 1.25 (± 0.46‰, n = 95) at RJ-1502. The δD-Alkenone values at RJ-1501 ranged from -181 and -159‰ (n = 47) and between -185 and -153‰ at RJ-1502 (n = 53) (Figure 3E). Decreasing values are observed during the deglaciation and increasing values during the early Holocene, with high values maintained (~159‰) up to ~ 5.5 ka, with a decreasing trend towards the present (up to ca. 1.7 ka).

## 4 Discussion

The synthesis of global SST carried out by the MARGO project inferred that the subtropical gyres in the Atlantic Ocean experienced very modest cooling (ca. 1 - 2 °C) in their center during the LGM (Waelbroeck et al., 2009). *G. ruber* Mg/Ca-derived SST estimates from core GL-1090 (24.92 °S, 42.51 °W, 2225 m)) agree with the MARGO compilation and did not indicate prominent cooling during the LGM (Santos et al., 2017a). Assuming a longer time-scale, the reported temperatures

from core GL-1090 during the LGM verified long-term warming developed since ca. 45 ka. According to Santos et al. (2017a), the absence of prominent cooling during the LGM and heat build-up in the region occurred in response to a progressively slower AMOC and glacial climate advance that stored warm waters in the subtropical South Atlantic gyre while sea-ice and ice-caps expanded in southern and northern high latitudes.

We compared the $U^{K'}_{37}$-derived SST from cores RJ-1501 and RJ-1502 to the late-glacial context proposed by Santos et al. (2017a) through core GL-1090. Figures 4A and B show that $U^{K'}_{37}$-derived SST was consistently colder than the Mg/Ca-derived SST throughout the last ca. 50 ka in the area. Despite the obvious deviations in terms of absolute temperature reconstruction that different ecology and calibration proxies will display, it is possible to address a similar pattern that emerges between the most offshore RJ-1502 and GL-1090. Both cores recorded progressive temperature increases since the late-glacial towards the Holocene, without significant cooling during the LGM (Figure 4A). This indicates that alkenones and surface-dwelling *G. ruber* were influenced by the BC core, which carried warm waters retained in the South Atlantic subtropical gyre to the Santos Basin during the end of the last glacial cycle. Interestingly, $U^{K'}_{37}$-derived SST from RJ-1501 followed an opposite trajectory to that indicated by RJ-1502 and GL-1090 (Figure 4B), where RJ-1501 presented a gradual surface cooling towards the LGM with accentuated warming only after ca. 18.5 ka, despite being separated by only 40 km from RJ-1502. The relative changes in ocean salinity indicated by δ¹⁸O *IVF-SW* agree with the patterns indicated by the SST reconstructions, i.e., RJ-1502 and GL-1090 cores displayed progressive ocean surface salinification from the late-glacial period to the Holocene (Figure 4C), while RJ-1501 presented surface freshening, achieving its maximum during the LGM and early-deglaciation (Figure 4D). The reconstructed core RJ-1501 parameters were only similar to those from RJ-1502 and GL-1090 during the Holocene (Figure 4). This is strong evidence that the planktic organisms accumulated in RJ-1501 core were under the influence of another oceanographic regime during the last glacial cycle.

One could argue that RJ-1501 presented a divergent evolution due to the influence of wind-driven shelf-break SACW upwelling. Although $U^{K'}_{37}$-derived SST has cooled continuously toward the LGM, temperatures were still warmer than those commonly reported for the occurrence of SACW shoaling (ca. 16 ºC) (Belem et al., 2013). Through the relative abundance of certain planktic foraminifera species, Lessa et al. (2017) proposed that upwelling in this region from the LGM to the Holocene was rather retracted because the glacial wind-regime (weak NE winds) did not favor the pumping of cold and nutrient-rich central waters to the photic zone. Portilho-Ramos et al. (2019) also consider a higher glacial productivity during earlier intervals of the last glacial (e.g., MIS 3) and from the LGM toward the Holocene the conditions for such productivity declined. Under this scenario, it could be considered that the Cabo Frio upwelling system was already confined to its modern position in inner parts of the continental shelf and far from core RJ-1501. Figure 3B exhibits the planktic *G. ruber* δ¹³C composition of cores RJ-1501 and RJ-1502. Several factors are recognized to influence the carbon isotope composition of planktic foraminifera, such as metabolic vital effects, the presence of symbiont bearing, the ocean carbonate chemistry and global distribution of ¹³C-depleted carbon between marine and terrestrial reservoirs (e.g., Bijma et al. 1998; Bijma, 2002; Spero et al., 1997; Oliver et al., 2010). The *G. ruber* δ¹³C reconstruction shows that RJ-1501 was slightly more depleted than RJ-1502 prior to ~22 ka, but from the LGM/last deglaciation interval both records present a very similar evolution with a pronounced

decline during the last deglaciation that has been suggested as a global imprint caused by the air-sea exchange (Lynch-Stieglitz et al., 2019). If RJ-1501 site were under the influence of strong upwelling of SACW during LGM and last deglaciation, it would be expected that the *G. ruber* $\delta^{13}$C of RJ-1501 would have shown a shift from the *G. ruber* $\delta^{13}$C of RJ-1502, at least in part, because of the influence of nutrient-enriched SACW compared to the nutrient-depleted surface Tropical Water (Kroopnick, 1985; Venancio et al., 2014). However, the good agreement of both *G. ruber* $\delta^{13}$C during the LGM and last deglaciation indicates that it is unlikely that cold and nutrient-rich tongues of SACW accounted for the surface water patterns seen in RJ-1501.

Once this alternative explanation is disregarded, it is remarkable that the temperature evolution of core RJ-1501 (with warming only after 18.5 ka) resembles the deglacial pattern inferred from mid- to high-latitudes records for the Southern Hemisphere. $U_{37}^{K'}$-derived SST from ODP Site 1233 (off southern Chile) presented deglacial warming initiated shortly after 19 ka (Figure 5A and B). According to the authors, such warming was a combined response of the temperature increase around Antarctica due to the bipolar seesaw during the Heinrich stadial 1 interval and the accentuated release of $CO_2$ from the deep ocean towards the atmosphere (Lamy et al., 2007) (Figure 5D and E). Mg/Ca-derived SST from core TNO57-21 (SE South Atlantic) indicated deglacial warming from ca. 18.5 ka, also in response to the pattern imposed by the bipolar seesaw mechanism (Barker et al., 2009). The mean air temperature reconstructed through lipids glycerol dialkyl glycerol tetraethers (GDGTs) from core GeoB6211-2 (SE Brazilian coast) displayed significant warming slightly after the records above (ca. 16.5 ka), but still within the main warming interval in Antarctica (Chiessi et al., 2015) (Figure 5C). A glaciolacustrine varved record (FCMC17) that report the dynamics of ice-margin retreat of the Patagonian Ice Sheet indicates that the varve thickness decreases in two-steps after the LGM, denouncing a prominent regional warming between 18 and 17.4 ka (Bendle et al., 2019). According to these authors, this glacial retreat highlights the potential synchronicity in atmospheric warming trends over the Southern Hemisphere mid- to high-latitudes with the onset of the last deglaciation. Part of the warming responsible for the Patagonian Ice Sheet melting recorded from 18 ka would be the result of unbalanced oceanic heat distribution due to the bipolar seesaw resulting in increasing temperatures in South Atlantic, South Pacific and Antarctica, with subsequent upwelling-driven $CO_2$ release from the Southern Ocean (EPICA, 2004; Lamy et al., 2007; Lüthi et al., 2008; Barker et al., 2009; Bendle et al., 2019) (Figure 5A-E). Putting these evidences together, such studies suggest that large areas of the northern portion of the Subtropical Front warmed during the early-last deglaciation after the LGM cooling relatively synchronous to Antarctica. It is reasonable to assume that the Malvinas Current, sourced from the northern limb of the Antarctic Circumpolar Current, could have transported this pattern northward, influencing Subantarctic Shelf Water formation. This temperature evolution characterized by a cold LGM and deglacial warming can be traced in lower latitudes along the SE Brazilian coast due to the northward-flowing BCC (Figure 5A-E). Apart from the impact of the BCC, cores RJ-1502 and GL-1090 (Santos et al., 2017a) did not record this mid- to high-Southern Hemisphere pattern (Figure 4A and B).

A recent investigation applying a dinocyst assemblage (core GeoB3202-1, 1090 m water depth) in a region near cores RJ-1501, RJ-1502, and GL-1090 also suggests the occurrence of cold SST during the LGM (Gu et al., 2019). These authors explain the discrepancy between their results and those reported by GL-1090 (Santos et al., 2017a) by invoking shifts in the

modern wind-driven upwelling area. Instead, we reason that such conflict could be reconciled by simply considering that shallower cores (RJ-1501, this study and GeoB3202-1, Gu et al., 2019) have a much higher chance of being within the BCC influence zone than deeper records (RJ-1502, this study and GL-1090, Santos et al., 2017a).

During the late-glacial period and Heinrich stadial 1 interval, it is very well documented that large areas in adjacent South America experienced wetter conditions due to a noticeable strengthening of the precipitation associated with the South America Monsoon System (SAMS) (Cruz et al., 2005; Stríkis et al., 2015; Novello et al., 2017; Gu et al., 2018; Stríkis et al., 2018). The increased precipitation volume would be drained by the Parana basin and subsequently discharged into the ocean by the La Plata River, mixing a continental freshwater influence with subantarctic shelf waters, similar to what occurs today (Burone et

al., 2013). In this context, BCC waters would carry not only a temperature pattern linked to the thermal evolution of the mid- to high-latitudes of the Southern Hemisphere, but also a sign of low salinity due to higher rainfall throughout the continent. The notion of a fresher surface over the shelf-break area is demonstrated by the $\delta^{18}O$ $_{IVF-SW}$ and $\delta D$ of RJ-1501 core, which became progressively lower throughout the late-glacial interval, producing a clear density contrast with the more saline offshore waters (Figure 5F and G). Hence, a stronger BCC influence would explain the residence of colder and fresher surface

waters over the RJ-1501 site (and other relatively shallow cores e.g., Gu et al. (2019)), but not in cores collected deeper from the continental slope. It is relevant to highlight that *G. ruber* $\delta^{18}O$ of RJ-1501 and RJ-1502 do not show significant differences throughout the studied period, except for the offset occurred in RJ-1501 to more negative values during the last deglaciation. This means that most of the $\delta^{18}O$ $_{IVF-SW}$ variability is being controlled by the $U^{K'}_{37}$-derived SST. Considering that in our interpretation, the $U^{K'}_{37}$-derived SST of RJ-1501 is sourced from southern areas of the continental shelf, the more substantial

influence of the SST upon the $\delta^{18}O$ $_{IVF-SW}$ constitutes as additional evidence to assume that RJ-1501 recorded a signal from higher latitudes of the western South Atlantic instead of that carried within the Tropical Water.

    In order to better demonstrate the contrast between the individual cores during the end of the last glacial cycle, we transferred the RJ-1501 age model to RJ-1502 and subtracted the average around zero (RJ-1502 minus RJ-1501) of $U^{K'}_{37}$-derived SST and $\delta^{18}O$ $_{IVF-SW}$ (i.e., $\Delta U^{K'}_{37}$-derived SST and $\Delta\delta^{18}O$ $_{IVF-SW}$) (Figure 6). The records were bootstrapped and 2.5[th] and 97.5[th] percentiles

are also presented (Figure 6). This exercise demonstrates that a sharp SST gradient (reaching >3ºC in its superior band) was formed toward the LGM and early-deglaciation, with a maximum between ca. 20 and 15 ka (Figure 6A). This gradient indicates that the BC offshore waters over the RJ-1502 site were ca. 2 ºC warmer than the inner waters over the RJ-1501 site, which is the double of the annual SST gradient observed today in the region. In the case of the $\Delta\delta^{18}O$ $_{IVF-SW}$, the difference achieves ca. 0.8 ‰ in the same period, corresponding to approximately a 1.5 salinity unit difference considering the seawater $\delta^{18}O$-salinity

relationship for the region (Belem et al., 2019).

    In regions where steep oceanic fronts develop, like the Southern Benguela where strong intrusions of warm Agulhas Current water occurs (Hardman-Mountford et al., 2003), the air-sea interaction works on a small scale and creates physical mechanisms of atmospheric response to temperature gradients of the sea surface that is reflected mainly over the wind stress and moisture anomalies (Saravanan and Chang, 2019). The across-shelf SST gradient caused by the shearing between the BC and BCC is

reported modulating the marine atmospheric boundary layer in the southern Brazilian continental shelf (Mendonça et al., 2017). The atmospheric turbulence caused by heat fluxes from the warm side of the ocean gradient has been investigated as an important factor influencing the weather of coastal regions off southern Brazil (de Camargo et al., 2013; Pezzi et al., 2016). Enhanced SST gradient is suggested as a factor that stimulates air-sea moisture exchange, which fuels annual coastal precipitation along with southern areas of the continental shelf (Reboita et al., 2010; Tirabassi et al., 2015). In a scenario of stronger northward spreading of BCC waters, like the one experienced during the LGM/early-deglaciation, the impacts over regional precipitation could extend to lower latitudes of the area. Therefore, our results demonstrate that the background conditions during the end of the last glacial cycle with a disturbed AMOC, reduced equatorial heat export and enhanced SAMS (leading to an increase in the La Plata river discharge) created conditions to accentuate the hydrographical contrast between the BC and BCC. Considering the future projections of AMOC weakening (Liu et al., 2017) and sea-level rising, the gradient strengthening reported herein may be a likely prospect for the southern Brazilian continental shelf.

## 5 Conclusions

In this study, we reconstructed the SST and $\delta^{18}O_{IVF-SW}$ through organic and inorganic geochemical proxies from two sediment cores collected from the upper (RJ-1501) and lower (RJ-1502) continental slope of the subtropical western South Atlantic. Although only 40 km apart, these records show considerably distinct hydrographic conditions throughout the end of the last glacial cycle. These contrasting results were reconciled assuming that the shallower and closer to the continent core RJ-1501 was under the influence of cold and fresh waters carried by the BCC, while the deeper and more offshore RJ-1502 was under the influence of the warm and saltier BC. A comparison with other records collected in the BC area supports this interpretation. Our results suggest that the conditions experienced during the last glacial transition, i.e., a weak AMOC and strong SAMS, increased the temperature and salinity gradient between the BC and BCC. Depending on the state of AMOC, this scenario may be accentuated in the coming decades.

## Data availability

Data of this study is available upon request to Letícia Luz (leticiagluz@gmail.com)

## Author Contributions

LL, TE and RS contributed to the study conception and design; LL and TS performed the age model calculation; LL, TS and RS wrote the first draft of the manuscript; LL, DM, BA, NH, SS and RN executed chemical analysis and/or contributed with sample preparation; All authors contributed to manuscript revision, read and approved the submitted version.

**Acknowledgments**

We thank Seaseep® and R/V Inspector II staff for sampling help. We also thank all members of Laboratory for Ion Beam Physicsat ETH Zurich for facilitating the [14]C measurements. We also thank the editor, Dr. Erin McClymont, and two anonymous reviewers which significantly contributed to improve the manuscript.

**Funding**

This study was financed in part by the Coordenação de Aperfeiçoamento de Pessoal de Nível Superior – Brasil (CAPES) – Finance Code 001, and by FAPERJ (proc. no. E-26/203.066/2017). Leticia Luz thanks CAPES for a doctorate (proc. no. 88881.151666/2017-00) and a PDSE (proc. no. 88881.134411/2016-01) scholarships. R. S. Carreira was supported by a research fellowship from CNPq (grant 309347/2017-3).

**Competing interest**

The authors declare that they have no conflict of interest.

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

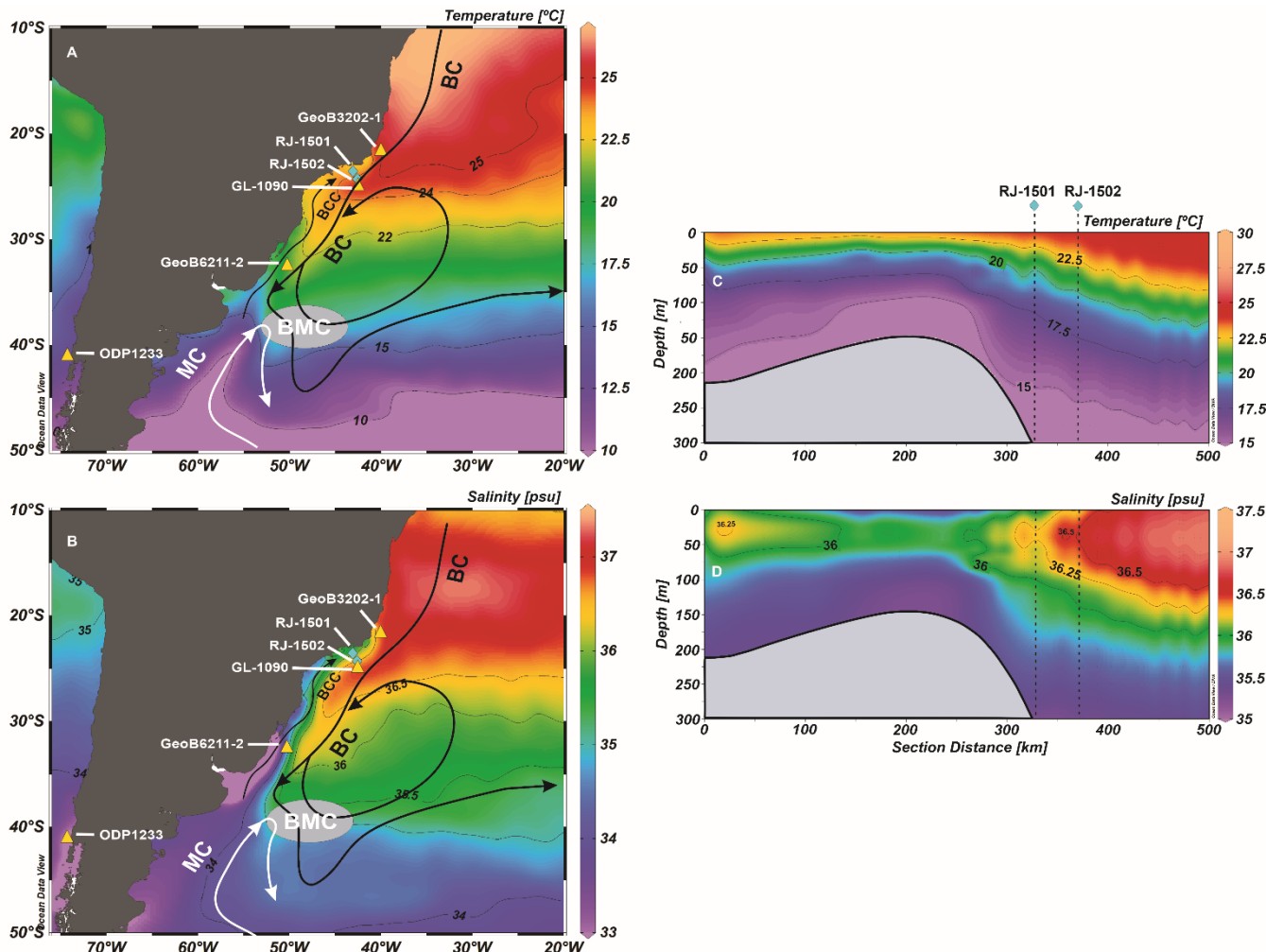

Figure 1 – Position of marine sediment cores RJ-1501 (23°58'14.3"S/43°06'35.1"W; 328 m water depth) and RJ-1502 (24°32'57.6" S/42°55'42.9"W; 1598 m water depth) in the upper and lower continental slope of the subtropical western South Atlantic and other cores discussed in this study. The maps feature the annual sea surface temperature (A) and sea surface salinity (B). The transects on panels (C) and (D) show the annual water mass structure relative to temperature and salinity, respectively. Acronyms on panels (A) and (B) define the Brazil Current (BC), Brazilian Coastal Current (BCC), Malvinas Current (MC), and Brazil-Malvinas Confluence (BMC). Temperature and salinity grids are derived from the World Ocean Atlas 2013 (Locarnini et al., 2013; Zweng et al., 2013). This figure was produced using the software Ocean Data View (Schlitzer, 2017).

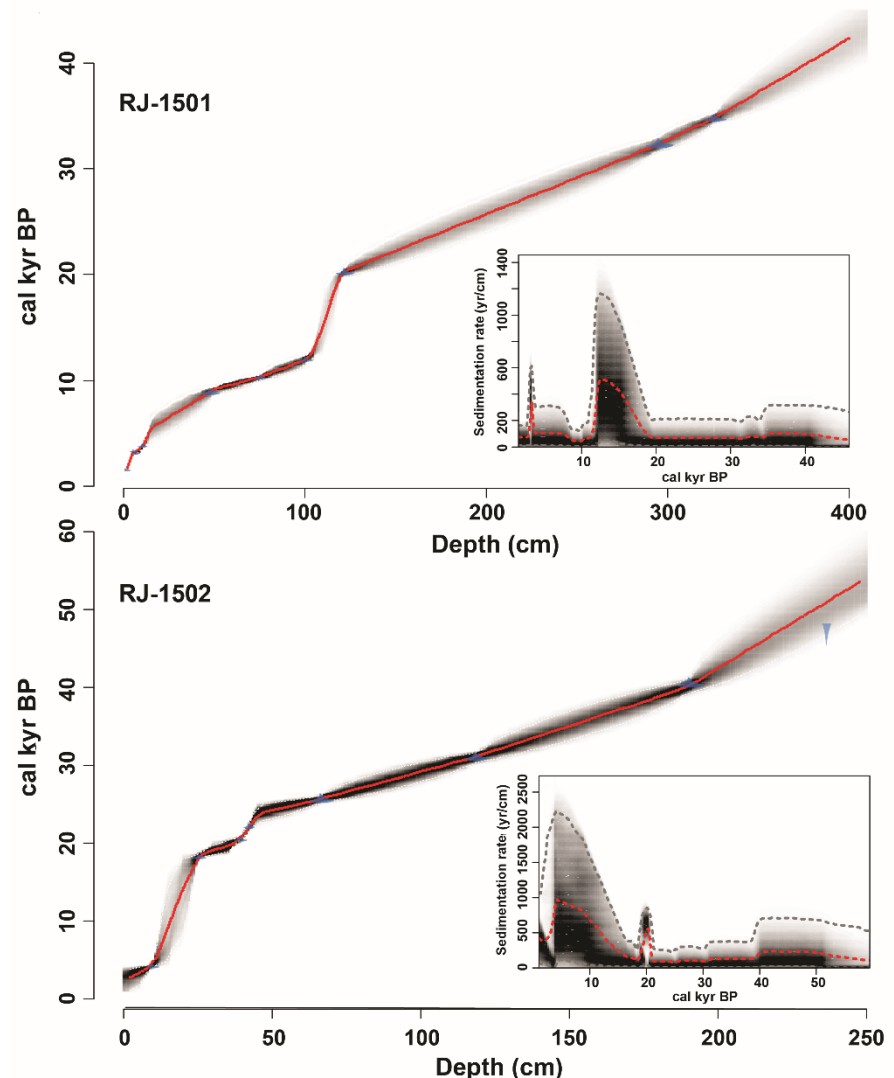

Figure 2 – Bayesian age-depth model and sedimentation rate (years/cm) for cores RJ-1501 (upper panel) and RJ-1502 (lower panel). The [14]Cages were calibrated with the curve IntCal13 (Reimer et al., 2013) and modeled with a reservoir age of 375 ± 36 years from ten local records. Thick (larger panels) and dashed (smaller panels) red lines depicted the highest probabilistic model for the ages and accumulation rate, respectively. Dashed (smaller panels) grey lines indicated the upper and bottom limits of the accumulation rate.

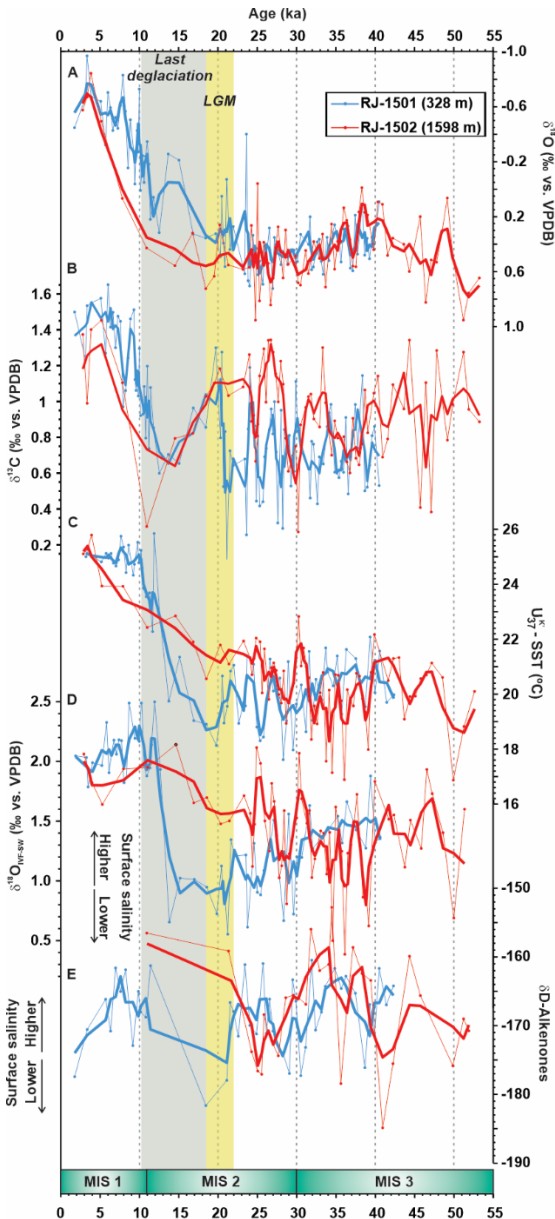

Figure 3 – Organic and inorganic proxies developed from marine sediment cores RJ-1501 (blue) and RJ-1502 (red). A: oxygen isotope ($\delta^{18}O$) of the planktic foraminifera *G. ruber*. B: carbon isotope ($\delta^{13}C$) of planktic foraminifera *G. ruber*. C: Alkenone ($U_{37}^{K'}$)-derived sea surface temperature (SST). D: Ice-volume free seawater oxygen isotope ($\delta^{18}O_{IVF-SW}$) derived from the $\delta^{18}O$ composition of *G. ruber* and $U_{37}^{K'}$-derived SST. E: $\delta D$-Alkenones. Records are depicted by the original data (dots and thin line) and the respective three-point running average (thick line). Yellow and grey bars denote the Last Glacial Maximum (LGM) and last deglaciation, respectively. Marine Isotope Stages (MIS) are indicated at the bottom of the panel.

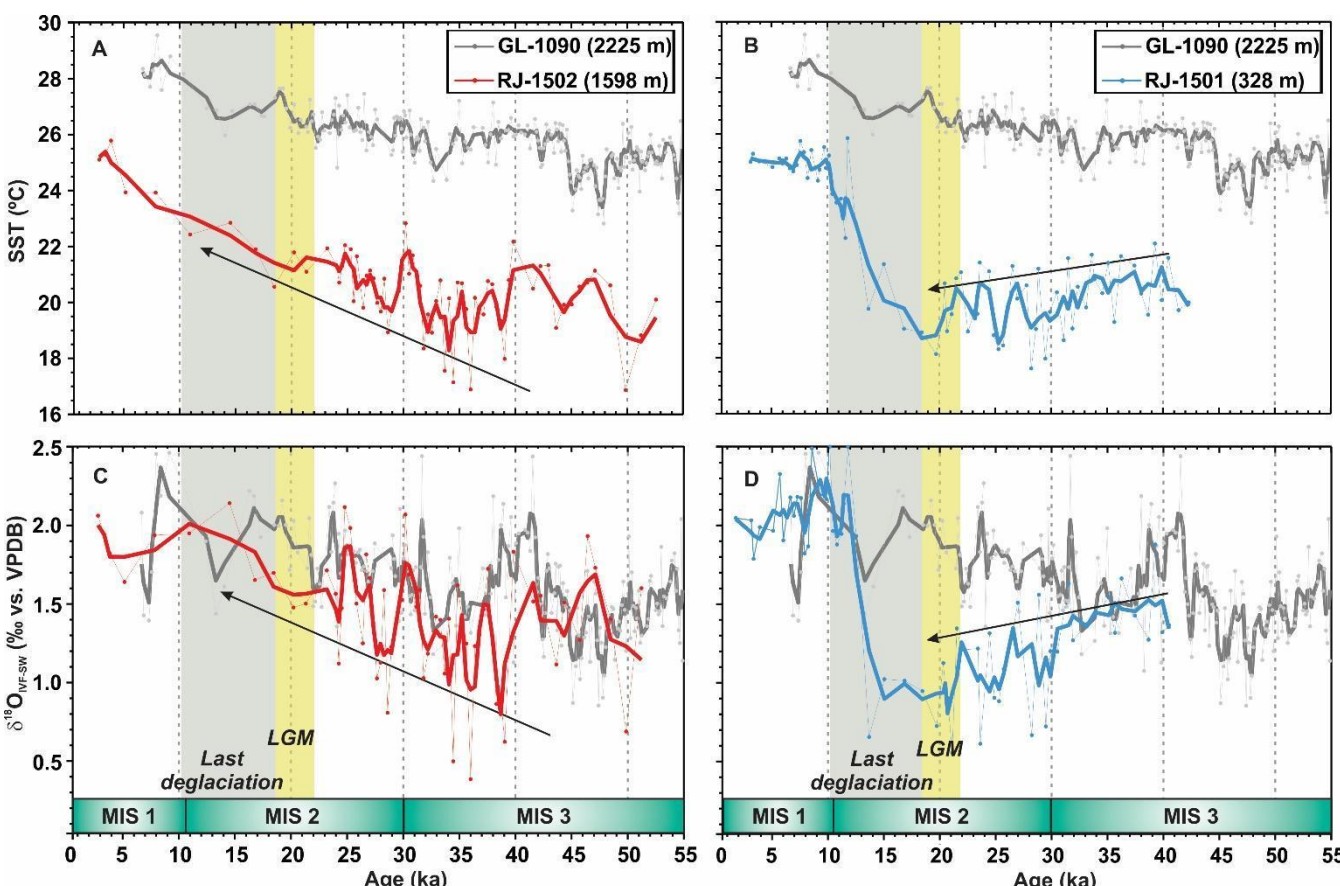

Figure 4 – Comparison of the records from marine sediment cores RJ-1501 (blue), RJ-1502 (red) (this study) and GL-1090 (grey) (Santos et al., 2017a). A and B: Alkenone ($U_{37}^{K'}$)-derived sea surface temperature (SST) from cores RJ-1502 and RJ-1501 and Mg/Ca-derived SST from core GL-1090. C and D: Ice-volume free seawater oxygen isotope ($\delta^{18}O_{IVF-SW}$) from cores RJ-1502, RJ-1501 and GL-1090. Records are depicted by the original data (dots and thin line) and the respective three-point running average (thick line). Yellow and grey bars denote the Last Glacial Maximum (LGM) and last deglaciation, respectively. Marine Isotope Stages (MIS) are indicated at the bottom of the panel.

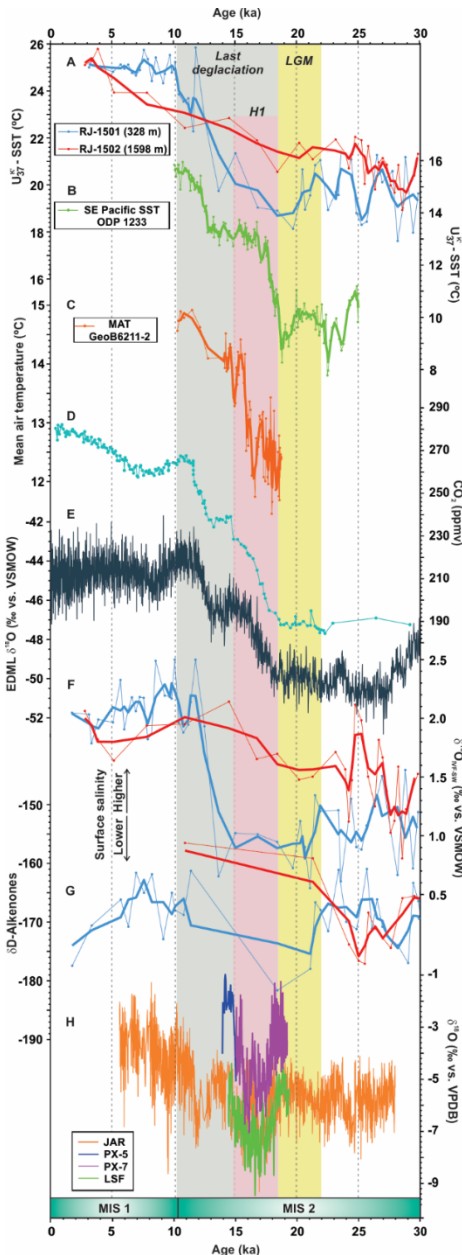

Figure 5 – Comparison of the marine sediment cores RJ-1501 (blue) and RJ-1502 (red) with other Southern Hemisphere records. A: Alkenone ($U^{K'}_{37}$)-derived sea surface temperature (SST) from cores RJ-1502 and RJ-1501. B: $U^{K'}_{37}$-derived SST from ODP Site 1233 (Lamy et al., 2007). C: Mean air temperature from GeoB6211-2 (Chiessi et al., 2015). D: Carbon dioxide ($CO_2$) concentration from Antarctic EPICA Dome C (Lüthi et al., 2008) on the Antarctic Ice Core Chronology (AICC2012) (Veres et al., 2013; Bazin et al., 2013). E: Antarctic oxygen isotope from EDML (EPICA, 2004) on the AICC2012 chronology. F: Ice-volume free seawater oxygen isotope ($\delta^{18}O_{IVF-SW}$) from RJ-1501 and RJ-1502. G: $\delta$D-Alkenones from RJ-1501 and RJ-1502. H: Speleothem oxygen isotope from Jaraguá cave (JAR) (Novello et al., 2017) and Paixão (PX-5 and PX-7) and Lapa sem Fim (LSF) caves (Stríkis et al., 2015). Records are depicted by the original data (dots and thin line) and the respective three-point running average (thick line). Yellow, red and grey bars denote the Last Glacial Maximum (LGM), Heinrich stadial 1 and last deglaciation, respectively Marine Isotope Stages (MIS) are indicated at the bottom of the panel.

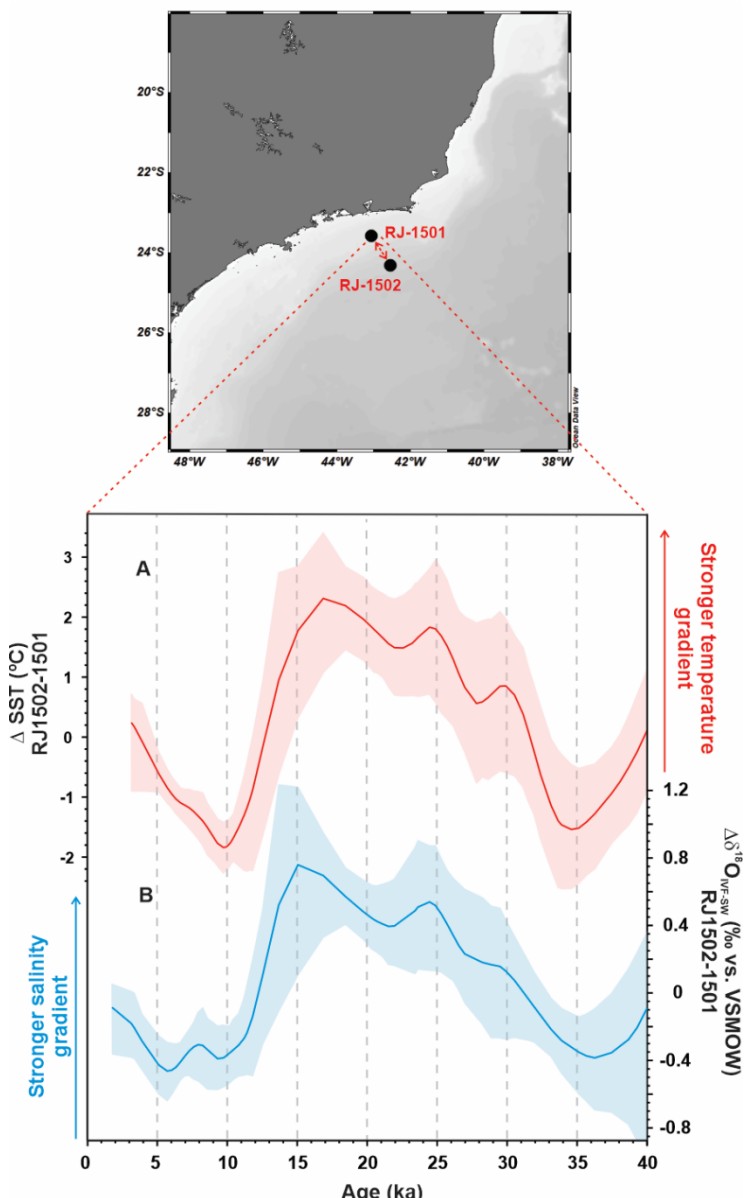

Figure 6 – Sea surface temperature and salinity gradient formed in the area between the marine sediment cores RJ-1501 and RJ-1502 (map in the uppermost panel) during the Last Glacial Maximum and early-deglaciation interval. Records were placed on a common time-scale (RJ-1501) and the mean around zero were subtracted (RJ-1502 minus RJ-1501) to produce ΔAlkenone ($U_{37}^{K'}$)-derived sea surface temperature (SST) and Δδ-ice-volume free seawater oxygen isotope ($\Delta\delta^{18}O_{IVF-SW}$). The records were bootstrapped using the Acycle software with a 10% window (Li et al., 2019) and presented with the 2.5th and 97.5th percentiles (red and blue shaded areas). A: $\Delta\delta U_{37}^{K'}$-derived SST. B: $\Delta\delta^{18}O_{IVF-SW}$.

**Table 1. Accelerator mass spectrometer radiocarbon (MICADAS) dates and calibrated ages used for age-depth models of cores RJ-1501 and RJ-1502.**

| Station | Core Depth (cm) | ID-Lab* | Species | Radiocarbon Age (yrBP) | ± 1s error | Calibrated Age (calyr BP) | Min-Max (calyr BP) |
|---------|-----|---------|---------|-----|-----|-----|-----|
| **RJ-1501** | 2 | 82195 | *G. ruber, T. sacculifer* | 2168 | 65 | 1756 | 1574-1930 |
| | 5 | 84479 | *G. ruber, T. sacculifer* | 3647 | 118 | 3138 | 2861-3397 |
| | 8 | 82194 | *G. ruber, T. sacculifer* | 3367 | 68 | 3361 | 3202-3558 |
| | 11 | 84478 | *G. ruber, T. sacculifer* | 4078 | 66 | 3904 | 3706-4094 |
| | 50 | 85107 | *G. ruber, T. sacculifer* | 8649 | 141 | 9145 | 8705-9496 |
| | 74 | 82193 | *G. ruber, T. sacculifer* | 9426 | 84 | 10273 | 9984-10533 |
| | 101 | 84477 | *G. ruber, T. sacculifer* | 10614 | 83 | 12090 | 11691-12453 |
| | 119 | 82192 | *G. ruber, T. sacculifer* | 17050 | 126 | 19706 | 19039-20272 |
| | 290 | 82191 | *G. ruber, T. sacculifer* | 28270 | 282 | 31955 | 31244-32779 |
| | 323 | 84476 | *G. ruber, T. sacculifer* | 30927 | 280 | 34585 | 34048-35143 |
| **RJ-1502** | 8 | 84509 | *G. ruber, T. sacculifer* | 4055 | 86 | 3841 | 3353-4239 |
| | 26 | 85049 | *G. ruber, T. sacculifer* | 15418 | 111 | 18454 | 18123-18793 |
| | 38 | 85048 | *G. ruber, T. sacculifer* | 16813 | 117 | 20236 | 19903-20614 |
| | 41 | 85047 | *G. ruber, T. sacculifer* | 18392 | 129 | 21321 | 20966-21676 |
| | 65 | 84510 | *G. ruber, T. sacculifer* | 21259 | 153 | 25523 | 24939-25966 |
| | 116 | 85106 | *G. ruber, T. sacculifer* | 26638 | 347 | 30784 | 29916-31447 |
| | 185 | 84508 | *G. ruber, T. sacculifer* | 34791 | 413 | 39395 | 38087-40532 |
| | 245 | 84507 | *G. ruber, T. sacculifer* | 49358 | 597 | 52887 | 48659-58741 |