# Peer review of "Contrasting late-glacial paleoceanographic evolution between the upper and lower continental slope of the western South Atlantic"

_Climate of the Past, 2020_

## Referee Comment (RC1) · Anonymous Referee #1 · 18 Feb 2020

General comments

This paper presents new records of organic and inorganic proxies, which are used to reconstruct changes in sea surface temperature and salinity off the Brazilian Margin over the last 50 kyr. The paper is well written and presents good interpretations. However, I think the authors need to clearly state their scientific questions and revise their approach regarding the salinity reconstruction (see my comments below). I am also not totally convinced that the authors can completely discard the influence of coastal upwelling in their records. Therefore, my recommendation is for a minor revision before publication in Climate of the Past.

[Figure]

Specific comments:

Abstract - Not clear by the first sentences what is the goal of the study. What is the scientific question? - All the proxies look traditional to me...what is the new proxy?

Introduction - Line 36: You start talking about millennial-scale changes, but finish the sentence with productivity changes citing papers that are not discussing millennial-scale mechanisms. This is a little bit confusing. Please consider revising the sentence. - Line 41: "application of cores"? Please consider revising this part. - Line 65: "Hence, BCC dynamics are a determining climate factor along the SE Brazilian coast". This is very vague; please explain the BCC dynamics and the influence of BCC on local climate. - Reading the entire introduction it remains unclear what is the main scientific question of the manuscript. The authors should make their goals very clear in the introduction.

Methods - Line 87: "Due to the chronological limitation of 14C dating, only the first 250 cm of the RJ-1502 core were considered in this study", but the core only has 250 cm as described in previous sentence. Did you analyze the entire core or not? - Changes in Fig 1: (i) add the main surface currents; (ii) include a figure with the water mass structure; (iii) consider expanding the map to include the La Plata River mouth and the other cores from this region that are mentioned in the discussion. - Fig 2: replace accumulation rates by sedimentation rates. - Line 161: The authors decided to use the equation of Müller et al. (1998). Why? What are the main arguments to use this particular equation and not the other available equations? - Please remove the first sentence of the topic 2.5. In which lab did you perform the d18O analysis? In which lab did you perform the delta-D analysis? - Line 194-196: The sentence is confusing. Please rewrite this sentence and better explain how you corrected the ice volume effect. - The authors use the SST derived from the alkenones in the paleotemperature equation of Mulitza et al. (2003). This is clearly not ideal and can generate errors. The authors should show arguments to support this approach. In my opinion, the ecology of these organisms from which the proxies are derived is very different (seasonality, habitat depth. . .), and

this is the reason why it is probably not right to use this approach.

Results - Line 199: Please write "Marine Isotope Stage" before using just the abbreviation "MIS" - Line 203: "G. ruber (white and pink)". But in the methods you said you just measured in G. ruber (white). Did you measure in both types?

Discussion - Line 231-233: The sentence is a little confusing. Please consider rewriting. When you say that the organisms present different ecologies, it gives support to my previous comment regarding the approach used for the d18Osw estimations. - Line 247: Temperatures below 20°C in the surface layer in this region can already indicate an influence of the SACW. - Line 248: The core used in Lessa et al. 2017 (GL-1090) is offshore. Core RJ-1501 is much more close to the coast and can be influenced by coastal upwelling. I think that the authors need to provide more arguments and show more data in order to complete discard the influence of coastal upwelling. - Line 274: Campos et al. 2019 have recently questioned this mechanism of intensification of the SAMS during Heinrich stadials. XRF data for cores south of 19°S show no increase in Ti/Ca or Fe/Ca (proxies for terrigenous input) during HS. The authors should include these recent findings in their discussion. - Line 293: It is not sufficient to just say "corresponding to almost 1.0 salinity unit (considering. . .". The authors need to show the SSS equation and the precise estimation of the salinity gradient.

Minor comments (technical corrections):

Line 17: remove "the" before salinity Line 22: signal from mid/high latitudes? The BCC do not come from so far south. Please consider rewriting the sentence. Line 25: ". . .which may have generated perturbations in the air-sea heat flux, with consequences for the regional SE South America climate". Could you be more specific? Line 32: remove "for" before several Line 36: replace "recovered" by "collected in the. . ." Line 66: Include "records from" before "sediment cores retrieved. . ." Line 76: I think it is better "without a minima in SST. . ." Line 125: Please use the more recent taxonomic nomenclature (Trilobatus sacculifer) Line 128: Write the full name of ETH. Line 141:

Please give the sedimentation rates (cm/yr) instead of the accumulation rates. Line 163: It seems that some words are missing in the end of the name of topic 2.5 Line 179: Please consider changing the name of the topic 2.6. Replace "tracers" by "proxies" and use the symbols of the proxies instead of the full name. Line 261: evidences instead of "evidence"

---

## Referee Comment (RC2) · Anonymous Referee #2 · 17 Mar 2020

Dear Leticia Luz,

It has been a pleasure reading your manuscript. I think it is a very interesting comparison between two datasets form two closely located cores. I think we can learn a lot form these kinds of studies including the one presented here. That being said, I have difficulties following the text. I have the feeling there is a lot of duplication in the description of the currents, for one and I think it would be really good if you would check the writing thoroughly again. On top of that there are some weird things I would like to mention, I doubt if anyone co-injected water with a known isotopic composition into a GC setup for alkenone analysis. I am guessing that the

nC27 n-alkane was not for quantification, you already describe quantification in the Uk section, but actually was used as isotope standard to be co-injected with your samples. The nC27 from Arndt (not Arna) Schimmelmann has a pre-determined isotopic composition. Hydrogen isotopes are expressed in ‰ relative to VSMOW (0‰. This complete mash-up of this methods section makes me wonder about the knowledge of the authors and the quality of measurements and/or the involvement or interest of the person that did the actual measurements? My slightly negative feelings are further strengthened by the ice volume free oxygen isotope record. According to the manuscript this was obtained by correcting for the Uk temperatures. So it is a temperature corrected ïĄď18O record, not and ice volume free ïĄď18O record? ïĄď18O of forams and I will ignore diagenetic overprinting, is determined by (calcification) temperature and the ïĄď18O of seawater. The latter is correlated with salinity and affected by ice volume especially in these glacial/interglacial records. To get to salinity the forma record has to be corrected for temperature and ice volume by subtracting a benthic foram record, for instance. If you did what you said, the IVF record does not only reflect changes in salinity? Be careful there. Your actual measured ïĄď18O records are not so different from each other, except maybe for the bump in the coastal record during the deglaciation. The temperature records are different and that basically determines the difference between the temperature corrected ïĄď18O records. Again, be careful with what you are looking at. In this case the temperature comes from different organisms than the ïĄď18O, which will result in additional uncertainties. The mismatch between the ïĄď2H of the alkenones and the ïĄď18O of the forams suggests that these organisms reflect different growth conditions, water masses and/or seasons which does not make it any easier. A Mg/Ca based temperature correction might be better. Of course, other people have also used Uk temperatures to correct ïĄď18O to get at water isotopic composition and with that salinity. So it is not necessarily wrong, just be careful and discuss this potential problem. Especially since your whole story is based on the temperature corrected ïĄď18O records and not the actual measured data. The last thing that makes me

wonder a little what is going on with this manuscript is the ïĄĎïĄď SST from figure 6, big delta as difference fine, little delta is for isotopes not Uk based SSTs. Very strange. All in all, I think that this is an interesting study, but I think the data needs a bit more work and I am not entirely sure the authors know exactly what they are doing or some of them have not seen the actual submitted version. As is it can not be published.

Please also note the supplement to this comment:
https://www.clim-past-discuss.net/cp-2020-4/cp-2020-4-RC2-supplement.pdf

————————————————————

---

## Author Comment (AC1) · 25 Apr 2020

Response to Reviewers of: *Contrasting late-glacial paleoceanographic evolution between the upper and lower continental slope of the western South Atlantic.*

We are grateful to the reviewers for their interest, attention to detail, and constructive comments that significantly improved the manuscript. Below, we respond to each of the reviewers' comments. We have copied the reviewers' comments in BLACK text and added our responses in BLUE text.

The revised manuscript is provided in the Word file labeled "Luz et al 2020_CPD_tracked changes" selected to examine our changes.

Leticia G. Luz (on behalf of the co-authors)

**Reviewer #1**

General comments

This paper presents new records of organic and inorganic proxies, which are used to reconstruct changes in sea surface temperature and salinity off the Brazilian Margin over the last 50 kyr. The paper is well written and presents good interpretations. However, I think the authors need to clearly state their scientific questions and revise their approach regarding the salinity reconstruction (see my comments below). I am also not totally convinced that the authors can completely discard the influence of coastal upwelling in their records. Therefore, my recommendation is for a minor revision before publication in Climate of the Past.

We are very grateful to Reviewer #1 for her/his detailed and constructive evaluation of our work. We think her/his suggestions helped to increase the manuscript quality. Below, we show how we addressed each of the points raised by Reviewer #1. At this point in the Response letter we wish to emphasize that we added in a figure with the *Globigerinoides ruber* $\delta^{13}$C results measured in cores RJ-1501 and RJ-1502. We think that this data is an additional support to discard an eventual strong upwelling of South Atlantic Central Water (SACW) at the RJ-1501 site during the Last Glacial Maximum (LGM) and last deglaciation. A discussion around these data was included in the correspondent part of the text and the potential limitation of planktic foraminifera $\delta^{13}$C was accounted for.

Specific comments

Abstract - Not clear by the first sentences what is the goal of the study. What is the scientific question? - All the proxies look traditional to me...what is the new proxy?

We removed the proxy classifications (traditional or new) as they have all been used for over 10 years even though some of them (cf. $\delta$D-Alkenones) are being considered for the first time in the study area. Additionally, we rephrased the beginning of the Abstract in order to better present the background and the main goal of our study, which is to improve the knowledge about the paleoceanographic evolution of shelf waters in the subtropical western South Atlantic.

Introduction

Line 36: You start talking about millennial-scale changes, but finish the sentence with productivity changes citing papers that are not discussing millennial scale mechanisms. This is a little bit confusing. Please consider revising the sentence.

The purpose of this first paragraph is to provide to the reader a general chronological overview of the studies along the Brazilian margin and their different approaches and conclusions depending on the sediment core region. That is the reason why we cited papers discussing millennial-scale mechanisms (common along NE Brazil) and finish with papers that do not discuss millennial-scale variability at all (those usually south of 20 ºS). We have rephrased the paragraph in order to make this clearer.

Line 41: "application of cores"? Please consider revising this part.

This sentence has been rephrased.

Line 65: "Hence, BCC dynamics are a determining climate factor along the SE Brazilian coast". This is very vague; please explain the BCC dynamics and the influence of BCC on local climate.

Part of the influence on local climate was mentioned in the previous statement where we present that "The gradient imposed by the front may disturb atmospheric properties such as surface wind stress, stability, and air-sea flux exchange because sea surface in this region can act as a heat source to the atmosphere". In order to present more arguments, we have considered the findings of the regional simulation by Reboita et al. (2010) showing that the SST gradient and the consequent air-sea exchange may influence the annual cycle of precipitation along the southern Brazilian continental shelf. In addition, other temperature gradients from other regions where the sea/atmosphere heat flux behave similarly to our study site were included in the beginning of the last paragraph from the Discussion section.

Reading the entire introduction it remains unclear what is the main scientific question of the manuscript. The authors should make their goals very clear in the introduction.

We agree with Reviewer #1 and added a statement at the end of the Introduction to emphasize the main goal of the study.

Methods

Line 87: "Due to the chronological limitation of 14C dating, only the first 250 cm of the RJ-1502 core were considered in this study", but the core only has 250 cm as described in previous sentence. Did you analyze the entire core or not?

The paragraph was rephrased to make it clear that the first 250 cm of the core RJ-1502 was used.

Changes in Fig 1: (i) add the main surface currents; (ii) include a figure with the water mass structure; (iii) consider expanding the map to include the La Plata River mouth and the other cores from this region that are mentioned in the discussion.

As required by Reviewer #1, we modified Figure 1 by adding the main surface currents, including additional panels to show the water mass structure and expanding the map to include La Plata River mouth. The locations of other cores mentioned in the manuscript were also added, except core TNO57-21, to preserve a map scale showing details of important hydrographic features.

Fig 2: replace accumulation rates by sedimentation rates.

The term "accumulation rate" was replaced by "sedimentation rates" in Figure 2 and in the text, accordingly.

Line 161: The authors decided to use the equation of Müller et al. (1998). Why? What are the main arguments to use this particular equation and not the other available equations?

Several calibration studies using core-top sediments proposed models to improve SST calculation from U37K´ (Conte et al., 2006; Müller et al., 1998; Tierney and Tingley, 2018 among other studies). Although these studies offer a global scope, some have a spotlight on a specific region. In 1998, Müller and co-authors analyzed alkenones in 149 surface sediments from the tropical to subpolar eastern South Atlantic to establish a sediment-based calibration of the U37K´ paleotemperature index. And so, the use of the equation of Müller et al. (1998) must take priority over all other equations in South Atlantic samples. Additionally, Ceccopieri et al., (2018) tested the equation of Müller et al (1998) to calculate the U37K´-derived SST values in the Campos Basin, in an area nearby to our cores, using both the annual mean core top calibration and based on the seasonal (austral) calibration curves. In this paper, the authors calculated the U37K´-derived SST data based on the calibrations of Müller et al (1998) and compared the results obtained with the World Ocean Atlas 2013 of SST (WOA13, Locarnini et al., 2013), concluding that the U37K´-derived SST agrees with the annual mean SST. Therefore, our decision to use the equation of Müller et al. (1998) is based on this previous published data. At the end of the last paragraph of item 2.4, we inserted a sentence to add contextual information.

Please remove the first sentence of the topic 2.5. In which lab did you perform the d18O analysis? In which lab did you perform the delta-D analysis?

We removed the first sentence of the topic 2.5 to improve the text of the paragraph. The delta-D analysis was performed at the Geological Institute (ETH-Zurich) and we added this information to the text.

Line 194-196: The sentence is confusing. Please rewrite this sentence and better explain how you corrected the ice volume effect.

This sentence has been rephrased.

The authors use the SST derived from the alkenones in the paleotemperature equation of Mulitza et al. (2003). This is clearly not ideal and can generate errors. The authors should show arguments to support this approach. In my opinion, the ecology of these organisms from which the proxies are derived is very different (seasonality, habitat depth. . .), and this is the reason why it is probably not right to use this approach.

In this new version of our manuscript, we have cited works in the Material and Methods section 2.6 that proceeded with the same assumption, that is, reconstruct the $\delta^{18}O_{IVF-SW}$ by combining planktic

foraminifera $\delta^{18}O$ and U37K´-derived SST (e.g., Rostek et al., 1993; Emeis et al., 2000; Carter et al., 2008; Sepulcre et al., 2011). Usually, some kind of seasonal correction was performed only on those studies that investigate areas with an extreme seasonal cycle in hydrographic parameters, as the Mediterranean Sea (Essallami et al., 2007). Ecological works suggest that *Globigerinoides ruber* and *Emiliania huxleyi* could be mostly found dwelling the same depth in the subtropical western South Atlantic (Venancio et al., 2017; Ceccopieri et al., 2018). We do not consider that the seasonal cycle in our area is drastic enough to shift our reconstruction to unlikely values. Furthermore, the $\delta^{18}O_{IVF-SW}$ reconstructed by our approach in core RJ-1502 and that of Santos et al. (2017) in core GL-1090 based only on planktic foraminifera show a reasonably good agreement in terms of absolute values and general trend throughout the last 50 ka (Figure 4C). This evidence suggests that our approach, although it has limitations associated with proxies uncertainty, is robust to reconstruct past variability of $\delta^{18}O_{IVF-SW}$.

**References (list relative to comments to reviewer 1 and 2)**

[revised manuscript text omitted]

---

## Author Comment (AC2) · 25 Apr 2020

Response to Reviewers of: *Contrasting late-glacial paleoceanographic evolution between the upper and lower continental slope of the western South Atlantic.*

We are grateful to the reviewers for their interest, attention to detail, and constructive comments that significantly improved the manuscript. Below, we respond to each of the reviewers' comments. We have copied the reviewers' comments in BLACK text and added our responses in BLUE text.

The revised manuscript is provided in the Word file labeled "Luz et al 2020_CPD_tracked changes" selected to examine our changes.

Leticia G. Luz (on behalf of the co-authors)

**Reviewer #2**

Here we try to subdivide the comment reviewer #2 in order to facilitate understanding of the response structure.

General comments

It has been a pleasure reading your manuscript. I think it is a very interesting comparison between two datasets form two closely located cores. I think we can learn a lot form these kinds of studies including the one presented here. That being said, I have difficulties following the text. I have the feeling there is a lot of duplication in the description of the currents, for one and I think it would be really good if you would check the writing thoroughly again.

We have tried to build the text to bring the arguments of our interpretation gradually to our reader. That is why in the discussion we (1) present the general findings of low-latitude climate during the last glacial for our studied region; (2) place the new records presented here in this context, highlighting the similarities and discrepancies among them; (3) exclude alternative explanations for the main divergence found in core RJ-1501 and; (4) present what we think is the most likely explanation for core RJ-1501 data in the light of Southern Hemisphere mid- to high-latitudes. We think that it is a very linear and logical sequence to follow. Regarding the regional settings (section 2.2), we could not find precisely (maybe citing line by line) where Reviewer #2 found the duplication in the description of the currents. The main water masses and currents influencing our region are always described by a unique designation, following some critical studies carried out in the region. Notwithstanding, in this new version, we present a broader map where the thermal and salinity gradients along the N-S transect are better visualized. A section map of the water masses also contributes to the oceanographic understanding. The position of cores discussed is now indicated.

Specific comments

On top of that there are some weird things I would like to mention, I doubt if anyone co-injected water with a known isotopic composition into a GC setup for alkenone analysis. I am guessing that the nC27 n-alkane was not for quantification, you already describe quantification in the Uk section,

but actually was used as isotope standard to be co-injected with your samples. The nC27 from Arndt (not Arna) Schimmelmann has a pre-determined isotopic composition. Hydrogen isotopes are expressed in ‰ relative to VSMOW (0‰. This complete mash-up of this methods section makes me wonder about the knowledge of the authors and the quality of measurements and/or the involvement or interest of the person that did the actual measurements?

We re-check the procedure and replaced the first paragraph of section 2.6 to clarify and add more detail to the methodology of the alkenones hydrogen isotopic composition, including where the samples were performed and the correcting the spelling of the $n$-C$_{27}$ standard's laboratory. We agree with Reviewer #2 and the $\delta^2$H quantification step for smaller and larger analytes amounts is more fully described in the text. In addition, we have also clarified that the samples were performed by the Timothy Eglinton team (ETH-Zurich) using the methodology of the hydrogen isotopic ratios of individual organic compounds applied to the previous studies (e.g., Makou et al., 2007; Häggi et al., 2019).

My slightly negative feelings are further strengthened by the ice volume free oxygen isotope record. According to the manuscript this was obtained by correcting for the Uk temperatures. So it is a temperature corrected $\delta^{18}$O record, not and ice volume free $\delta^{18}$O record? $\delta^{18}$O of forams and I will ignore diagenetic overprinting, is determined by (calcification) temperature and the $\delta^{18}$O of seawater. The latter is correlated with salinity and affected by ice volume especially in these glacial/interglacial records. To get to salinity the forma record has to be corrected for temperature and ice volume by subtracting a benthic foram record, for instance. If you did what you said, the IVF record does not only reflect changes in salinity? Be careful there. Your actual measured $\delta^{18}$O records are not so different from each other, except maybe for the bump in the coastal record during the deglaciation. The temperature records are different and that basically determines the difference between the temperature corrected $\delta^{18}$O records. Again, be careful with what you are looking at. In this case the temperature comes from different organisms than the $\delta^{18}$O, which will result in additional uncertainties. The mismatch between the $\delta^2$H of the alkenones and the $\delta^{18}$O of the forams suggests that these organisms reflect different growth conditions, water masses and/or seasons which does not make it any easier. A Mg/Ca based temperature correction might be better. Of course, other people have also used Uk temperatures to correct $\delta^{18}$O to get at water isotopic composition and with that salinity. So it is not necessarily wrong, just be careful and discuss this potential problem. Especially since your whole story is based on the temperature corrected $\delta^{18}$O records and not the actual measured data.

We agree with Reviewer #2 that the full details regarding the ice-volume free seawater $\delta^{18}$O ($\delta^{18}$O$_{\text{IVF-sw}}$) was not properly described in the original submission. We accounted for that in this new version by adding more explanations of how the records were produced. It is important to emphasize that the $\delta^{18}$O$_{\text{IVF-SW}}$ is not a temperature-corrected record but indeed a $\delta^{18}$O corrected record. The sea-level/ice-volume correction, in this case, is assumed from Grant et al. (2012), and the meters of sea-level change is translated to their equivalents in seawater $\delta^{18}$O considering a glacial $\delta^{18}$O-enrichment of 0.008 ‰ per meter sea-level decay (Schrag et al., 2002). The fact that the *G. ruber* $\delta^{18}$O is not so different from each other, except maybe for the bump in the coastal record during the deglaciation, is the central pillar of our argumentation. The offset (bump) noted by Reviewer #2 from the LGM to the last deglaciation was likely caused by the intrusion of fresh coastal waters flowing from the southern shelf. In the new section 2.5, we deal with the eventual bias that could be generated by applying two different organisms to reconstruct $\delta^{18}$O$_{\text{IVF-SW}}$. We cited other references

that have done the same on the grounds that the depth habitat of *Emiliania huxleyi*, the dominant alkenone producer, and *G. ruber* is comparable (e.g., Rostek et al., 1993; Emeis et al., 2000; Carter et al., 2008; Sepulcre et al., 2011), which is also the case of the subtropical western South Atlantic (Venancio et al., 2017; Ceccopieri et al., 2018). Seasonal corrections over the U37K´-derived SST before application in $\delta^{18}O_{IVF-SW}$ has been used only in regions of extreme seasonal variations in temperature and salinity, as the Mediterranean Sea (e.g., Essallami et al., 2007), which is not the case of the subtropical western South Atlantic. $\delta^{18}O_{IVF-SW}$ and $\delta D$ are not conflicting since both are showing that RJ-1501 suffered the influence of a fresher surface water. It is worthy to note the comparison of $\delta^{18}O_{IVF-SW}$ between RJ-1502 (the most offshore record of our study) and that of GL-1090 (Santos et al., 2017) presented in Figure 4C. The foraminifera-only $\delta^{18}O_{IVF-SW}$ of GL-1090 and the alkenone-foraminifera $\delta^{18}O_{IVF-SW}$ of RJ-1502 are rather similar in terms of general trend and values. If some kind of strong bias because of ecology preferences was taking place the signals would be separated by large offsets, which is not the case. Figure 4C shows that, at the end, the hydrographic features in which the organisms are exposed is likely more important than their biological singularities. Indeed, for standardization proposes a foraminifera-only $\delta^{18}O_{IVF-SW}$ would be the best scenario, but unfortunately, producing a *G. ruber* Mg/Ca at this point is a suggestion impossible to overcome. The samples presented here were analyzed at ETH (Switzerland) and there is no financial and logistical support for this to be repeated (as a result of the troubled moment that Brazilian science lives added to the impacts of Covid-19). Furthermore, the analytical routine for Mg/Ca is not yet implemented in Brazil.

The last thing that makes me wonder a little what is going on with this manuscript is the $\Delta\delta$ SST from figure 6, big delta as difference fine, little delta is for isotopes not Uk based SSTs. Very strange. All in all, I think that this is an interesting study, but I think the data needs a bit more work and I am not entirely sure the authors know exactly what they are doing or some of them have not seen the actual submitted version. As is it can not be published.

We have used the notation as "$\Delta\delta$" because we are doing a double subtraction of the SST. The first "$\delta$" would come from the subtraction of the mean around zero (anomaly) of each record by itself. The second "$\Delta$" would come from the subtraction of the mean around zero between the records placed on a common timescale. We agree that this may cause confusion and, in this new version, we adopted only the single "$\Delta$" notation. Once more, it would be useful if Reviewer #2 could indicate by line, paragraph or section where he/she thinks the data needs a bit more work (as was the case of Reviewer #1). We respectfully would like to emphasize that all coauthors have the opportunity to see the manuscript before the submission and any statement opposed to that is just speculation.

**References (list relative to comments to reviewer 1 and 2)**

Carter, L., Manighetti, B., Ganssen, G. and Northcote, L.: Southwest Pacific modulation of abrupt climate change during the Antarctic Cold Reversal-Younger Dryas, Palaeogeogr. Palaeoclimatol. Palaeoecol., 260(1–2), 284–298, doi:10.1016/j.palaeo.2007.08.013, 2008.

Ceccopieri, M., Carreira, R. S., Wagener, A. L. R., Hefter, J. H. and Mollenhauer, G.: On the application of alkenone- and GDGT-based temperature proxies in the south-eastern Brazilian continental margin, Org. Geochem., 126, 43–56, doi:https://doi.org/10.1016/j.orggeochem.2018.10.009, 2018.

Conte, M. H., Sicre, M.-A., Rühlemann, C., Weber, J. C., Schulte, S., Schulz-Bull, D. and Blanz, T.: Global temperature calibration of the alkenone unsaturation index (UK′37) in surface waters and comparison with surface sediments, Geochemistry, Geophys. Geosystems, 7(2), doi:10.1029/2005GC001054, 2006.

Emeis, K.-C., Struck, U., Schulz, H.-M., Rosenberg, R., Bernasconi, S., Erlenkeuser, H., Sakamoto, T. and Martinez-Ruiz, F.: Temperature and salinity variations of Mediterranean Sea surface waters over the last 16,000 years from records of planktonic stable oxygen isotopes and alkenone unsaturation ratios, Palaeogeogr. Palaeoclimatol. Palaeoecol., 158(3–4), 259–280, doi:10.1016/S0031-0182(00)00053-5, 2000.

Essallami, L., Sicre, M. A., Kallel, N., Labeyrie, L. and Siani, G.: Hydrological changes in the Mediterranean Sea over the last 30,000 years, Geochemistry, Geophys. Geosystems, 8(7), doi:10.1029/2007GC001587, 2007.

Grant, K. M., Rohling, E. J., Bar-Matthews, M., Ayalon, A., Medina-Elizalde, M., Ramsey, C. B., Satow, C. and Roberts, A. P.: Rapid coupling between ice volume and polar temperature over the past 150,000 years, Nature, 491(7426), 744–747, doi:10.1038/nature11593, 2012.

Häggi, C., Eglinton, T. I., Zech, W., Sosin, P. and Zech, R.: A 250 ka leaf-wax δD record from a loess section in Darai Kalon, Southern Tajikistan, Quat. Sci. Rev., 208, 118–128, doi:https://doi.org/10.1016/j.quascirev.2019.01.019, 2019.

Locarnini, R. A., Mishonov, A. V, Antonov, J. I., Boyer, T. P., Garcia, H. E., Baranova, O. K., Zweng, M. M., Paver, C. R., Reagan, J. R., Johnson, D. R., Hamilton, M., Seidov, D. and Technical: World Ocean Atlas 2013, edited by S. Levitus and M. A., NOAA Atlas NESDIS 73., 2013.

Makou, M. C., Hughen, K. A., Xu, L., Sylva, S. P. and Eglinton, T. I.: Isotopic records of tropical vegetation and climate change from terrestrial vascular plant biomarkers preserved in Cariaco Basin sediments, Org. Geochem., 38(10), 1680–1691, doi:http://dx.doi.org/10.1016/j.orggeochem.2007.06.003, 2007.

Müller, P. J., Kirst, G., Ruhland, G., von Storch, I. and Rosell-Melé, A.: Calibration of the alkenone paleotemperature index U37K′ based on core-tops from the eastern South Atlantic and the global ocean (60°N-60°S), Geochim. Cosmochim. Acta, 62(10), 1757–1772, doi:https://doi.org/10.1016/S0016-7037(98)00097-0, 1998.

Reboita, M., Rocha, R., Ambrizzi, T. and Caetano, E.: An assessment of the latent and sensible heat flux on the simulated regional climate over Southwestern South Atlantic Ocean, Clim. Dyn., 34, 873–889, doi:10.1007/s00382-009-0681-x, 2010.

Rostek, F., Ruhland, G., Bassinot, F., Muller, P., Labeyrie, L., Lancelot, Y. and Bard, E.: Reconstructing sea surface temperature and salinity using d18O and alkenone records, Nature, 364, 319–321, 1993.

Santos, T. P., Lessa, D. O., Venancio, I. M., Chiessi, C. M., Mulitza, S., Kuhnert, H., Govin, A., Machado, T., Costa, K. B., Toledo, F., Dias, B. B. and Albuquerque, A. L. S.: Prolonged warming of the Brazil Current precedes deglaciations, Earth Planet. Sci. Lett., 463, 1–12, doi:https://doi.org/10.1016/j.epsl.2017.01.014, 2017.

Schrag, D. P., Adkins, J. F., McIntyre, K., Alexander, J. L., Hodell, D. A., Charles, C. D. and McManus, J. F.: The oxygen isotopic composition of seawater during the Last Glacial Maximum, Quat. Sci. Rev., 21(1–3), 331–342, doi:10.1016/S0277-3791(01)00110-X, 2002.

Sepulcre, S., Vidal, L., Tachikawa, K., Rostek, F. and Bard, E.: Sea-surface salinity variations in the northern Caribbean Sea across the Mid-Pleistocene Transition, Clim. Past, 7(1), 75–90, doi:10.5194/cp-7-75-2011, 2011.

Tierney, J. E. and Tingley, M. P.: BAYSPLINE: A New Calibration for the Alkenone Paleothermometer, Paleoceanogr. Paleoclimatology, 33(3), 281–301, doi:10.1002/2017PA003201, 2018.

Venancio, I. M., Belem, A. L., Santos, T. P., Lessa, D. O. and Albuquerque, A. L. S.: Calcification depths of planktonic foraminifera from the southwestern Atlantic derived from oxygen isotope analyses of a four - year sediment trap series, Mar. Micropaleontol., 136(August), 37–50, doi:10.1016/j.marmicro.2017.08.006, 2017.

---

## Author Response (ED1)

Response to the Editor on comments on R1 manuscript entitled: *Contrasting late-glacial paleoceanographic evolution between the upper and lower continental slope of the western South Atlantic*.

Dear Dr Erin McClymont

We are grateful for your reply with comments on our revised manuscript. The new comments on the revision #1 manuscript were taken in consideration and all suggestions incorporated in this revised manuscript, as detailed below.

Leticia G. Luz (on behalf of the co-authors)

**Editor comments**

15    Reviewer 2 commented on some duplication of information about the circulation. In the revised manuscript I can see where this arises – it is not word-for-word duplication, but you have a detailed discussion at the start of page 3 about the different water masses in this region and how they interact, which then feeds into how different previous work might have found differing results. I can understand the rationale for providing this information, since it gives a good justification for the study. But when I

20    was reading this I thought "this would make a good section on it's own", and yet in Section 2 there is a section called "regional settings" which also talks about circulation patterns and their controls in more depth. I recommend that the authors consider shortening the introduction e.g. by finishing page 2 with a statement of the study aims and then moving onto a "regional setting" section (having used Figure 1 to flag that there is complex circulation). Or, if the Introduction stays as it is, section 2.2 could be more

25    focussed on the direct controls over circulation at these two sites (e.g. section 2.2 describes processes happening at 10-15*S, at 33-38*S etc but some of this is visible in Figure 1).

*R. We decided to reduce the Introduction section and to give more details about the characteristics of the BCC in the Regional Settings section*

The final paragraph of the Introduction includes text (from "our multi-proxy reconstruction indicates..." which explains the findings of the study. I recommend to remove this text which is "Discussion" to keep the focus of the Introduction on the rationale and aims for the study and then what was done.

*R. The text mentioned was removed. We only kept the last sentence, because we believe it illustrates a broad context in which our work is inserted.*

I recommend swapping #2.1 and #2.2 so that the information on the sediment cores is directly adjacent to the age model information (#2.3).

*R. OK, the change is performed.*

page 4 lines 3-4: place the information about selecting the top 250 cm into it's own sentence rather than in brackets.

*R. OK*

You discuss the uncertainty on the d18OIVF-SW results. Could you please flag these in the caption to Figure 3 (and Figure 4), so that it is clear that the variations you are calculating exceed those propagated uncertainties? (even better – show a vertical bar for the size of the uncertainties if you can't show them on each data point, but I can see that you don't have much space on this graphic). It could be particularly valuable in Figure 4 to show that both data sets 'overlap' or not.

*R. We added a vertical bar in Figs 3 and 4 and this information now appears in the figures' captions. These bars represent the total uncertainty associated with the $^{18}O_{ivf-sw}$ reconstruction. It is noteworthy t that even considering the maximum uncertainty in the measure, there are no overlaps in our records during the period of time focused on our manuscript, and hence the proxy changes observed in the cores represent a true climatic signal and not an artifact derived from measure uncertainties.*

page 8 line 16: some duplication in the text ("the 101" is stated twice)

*R. OK*

In Discussion (p.10 and Figure 5), where the regional signals are compared, it would also be good to consider the record of Roberts et al. (https://www.sciencedirect.com/science/article/pii/S0012821X17303825) from the south west Atlantic. Roberts showed alkenone SST changes upstream of the Drake Passage, and made comparisons between the Santos data cited here as well as data from the Chilean margin, including observations of anti-phase SST patterns (rather than synchroneity noted here). The Roberts paper discusses circulation implications for the south west Atlantic so could provide some interesting comparison.

75

*R. The data from the Roberts et al paper (2007) are incorporated in the new version of the manuscript and a brief discussion based on the paper is included in the caption of Figure 5. In addition, the data from the Chilean margin (MD07-2138; Canujapan et al. 2011) are also included in this figure, because these authors also discussed the data of Roberts et al (2007). Accordingly, Figure 1 now also show these*
80    *new cores (CG528 and MD07-3128). The data from the core CG528 corroborate with our interpretation of the relatively higher temperatures during the LGM based on the RJ-1502 data (and the GL-1090). However, we point out that the colder SST temperatures registered in the CG528 along the last deglaciation is not consistent with our data as well as other data from the same period obtained in cores from the Pacific Ocean.*

[revised manuscript text omitted]

---

## Author Response (AR3)

Response to the Editor on comments on R2 manuscript entitled: *Contrasting late-glacial paleoceanographic evolution between the upper and lower continental slope of the western South Atlantic*.

Dear Dr Erin McClymont

Herein is the new revised version of the manuscript which addresses the technical issues you raised in your last feedback.

Thank you for critical reading and suggestions during manuscript preparation.

Sincerely,

Leticia G. Luz (on behalf of the co-authors)

[revised manuscript text omitted]